# Water vapor increase in the lower stratosphere of the Northern Hemisphere due to the Asian monsoon anticyclone observed during the TACTS/ESMVal campaigns

Christian Rolf[1], Bärbel Vogel[1], Peter Hoor[2], Armin Afchine[1], Gebhard Günther[1], Martina Krämer[1], Rolf Müller[1], Stefan Müller[2,3], Nicole Spelten[1], and Martin Riese[1]

[1]Institute for Energy and Climate Research (IEK-7), Forschungszentrum Jülich GmbH, Jülich, Germany.
[2]Institute for Atmospheric Physics, Johannes Gutenberg University, Mainz, Germany.
[3]now at: Enviscope GmbH, 60489 Frankfurt, Germany.

*Correspondence to:* C. Rolf (c.rolf@fz-juelich.de)

**Abstract.** The impact of air masses originating in Asia and influenced by the Asian monsoon anticyclone on the Northern Hemisphere stratosphere is investigated based on in situ measurements. A statistically significant increase in water vapor ($H_2O$) of about 0.5 ppmv (11 %) and methane ($CH_4$) of up to 20 ppbv (1.2 %) in the extra-tropical stratosphere above a potential temperature of 380 K was detected between August and September 2012 during the HALO aircraft missions Transport and Composition in the UT/LMS (TACTS) and Earth System Model Validation (ESMVal). We investigate the origin of the increased water vapor and methane using the three-dimensional Chemical Lagrangian Model of the Stratosphere (CLaMS). We assign the source of the moist air masses in the Asian region (North and South India, East China, South East Asia, and the tropical Pacific) based on tracers of air mass origin used in CLaMS. The water vapor increase is correlated with an increase of the simulated Asian monsoon air mass contribution from about 10 % in August to about 20 % in September, which corresponds to a doubling of the influence from the Asian monsoon region. Additionally, back trajectories starting at the aircraft flight paths are used to differentiate transport from the Asian monsoon anticyclone and other source regions by calculating the Lagrangian cold point (LCP). The geographic location of the LCPs, which indicates the region where the imprint of water vapor mixing ratio along these trajectories occurs, can be predominantly attributed to the Asian monsoon region.

## 1 Introduction

Radiatively active trace gases, such as water vapor and methane, play a key role in determining the radiative balance in the upper troposphere and lower stratosphere (UTLS) and thus have an impact on the surface climate of the Earth (Forster and Shine, 2002; Riese et al., 2012). In particular, only small increases/decreases of stratospheric water vapor in the order of 10 % have a significant influence on the radiative forcing and thus constitute a warming/cooling potential for global surface temperature in the order of 25-30 % (Solomon et al., 2010). In this paper, we focus on transport pathways and exchange processes between troposphere and stratosphere to better understand the variability of water vapor and methane in this climatically sensitive region of the atmosphere.

In general, freeze-dried air masses with a low amount of water vapor enter the stratosphere in the tropical tropopause layer (TTL) and are transported vertically via the Brewer–Dobson circulation (BDC) deep into the stratosphere and quasi-horizontally into the extra-tropical lower stratosphere (Ex-LS) (e.g., Gettelman et al., 2011). Thus, the air in the Ex-LS constitutes a mixture of young air masses originating from quasi-horizontal transport out of the TTL and old stratospheric air in the downwelling branch of the BDC (e.g., Bönisch et al., 2009; Ploeger et al., 2013; Vogel et al., 2016). As a rule, an increasing inflow of tropospheric air masses from the tropics in combination with higher temperatures in the tropopause region causes a moistening of the Ex-LS in summer compared to winter months (e.g., Hoor et al., 2005; Krebsbach et al., 2006; Ploeger et al., 2013; Randel and Jensen, 2013; Zahn et al., 2014). This moistening is partly due to the Asian summer monsoon which facilitates the transport of water vapor into the northern lowermost stratosphere (e.g., Dethof et al., 1999; Randel and Jensen, 2013). The monsoon system is linked to rapid vertical transport of surface air from India and east Asia in the summer months (July to October). These young and moist tropospheric air masses from close to the surface are to some degree horizontally confined in the upper troposphere by the Asian monsoon anticyclone (AMA). The edge of the AMA constitutes a transport barrier characterized by a potential vorticity (PV) gradient (Ploeger et al., 2015). The transport barrier is rather leaky due to the strong dynamic variability of the anticyclone consisting of meridional displacements, splits, and eddy-shedding events (e.g., Vogel et al., 2014, 2015; Ungermann et al., 2016). However, this confinement with reduced cross-gradient transport is responsible for an enrichment of tropospheric constituents and higher amounts of water vapor or methane in the upper troposphere (e.g., Park et al., 2004, 2007; Schuck et al., 2010; Patra et al., 2011; Baker et al., 2012; Bian et al., 2012; Pan et al., 2016; Santee et al., 2017). A horizontal transport pathway out of the anticyclone is directed westwards into the tropical tropopause layer (TTL) (Popovic and Plumb, 2001; Randel and Park, 2006). A second pathway of moist and polluted air masses exists, where air from the AMA is transported eastwards along the subtropical jet and subsequently into the northern lower stratosphere via eddy shedding followed by tropopause crossing (e.g., Dethof et al., 1999; Hsu and Plumb, 2000; Vogel et al., 2014, 2016). These frequently occurring eddy-shedding events triggered by Rossby wave activity flood the northern lower stratosphere with tropospheric air masses from the Asian subcontinent and cause a moistening of the Ex-LS (e.g., Müller et al., 2016; Vogel et al., 2014, 2016). In addition, Aura Microwave Limb Sounder (MLS) satellite observations showed that the Asian monsoon is hydrating the northern stratosphere (Uma et al., 2014).

From an observational point of view, the influence of the Asian monsoon on the Ex-LS is mostly analyzed based on satellite limb sounder measurements, which only have a limited vertical resolution, typically larger than 1-2 km in the lowermost stratosphere (e.g., Hegglin et al., 2013; Santee et al., 2017). Especially in the tropopause region, this coarse resolution can smooth the strong vertical gradient of water vapor at the tropopause and may lead to both an over- or underestimation of the water vapor concentration in the lower stratosphere. Here, we present high-resolution and precise in situ water vapor and methane measurements in the northern lower stratosphere performed in August and September 2012. We investigate the changes of these trace gases during the observational time period and attribute them to transport from the Asian monsoon region by using Lagrangian back-trajectory analysis in combination with artificial tracers of air mass origin.

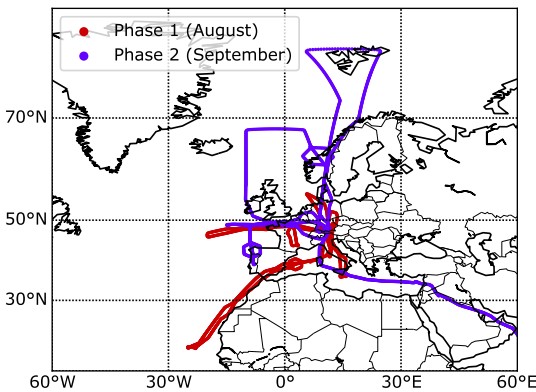

**Figure 1.** Flight paths of all analyzed flights during the TACTS and ESMVal campaigns. The flight paths of phase 1 and phase 2 are plotted in red and blue, respectively.

## 2    Methods and Instruments

This study is based on the data collected during the two aircraft campaigns Transport and Composition in the UT/LMS (TACTS) and Earth System Model Validation (ESMVal). These campaigns were conducted between August and October 2012 mainly in the Northern Hemisphere in the region from the Cape Verde Islands to the Arctic and over Europe and the Atlantic Ocean. The data are split into two time periods. These two periods, referred to as phase 1 and phase 2 hereafter, cover the time period from August 28 to September 5 and from September 18 until September 26, 2012, respectively. Figure 1 shows the map of all considered flight paths and the respective phases they are attributed to. Vogel et al. (2016) provided the motivation for this selection, showing that the strongest transport of young air masses from the AMA into the Ex-LS occurred in the period between August and the end of September, mainly associated with a pronounced eddy-shedding event on 20 September 2012 (Vogel et al., 2014). Thus, the selection of these dates reveals a clear difference in the water vapor distribution in the Ex-LS, as shown in Sec. 3.1. More information about the data coverage can be found in Müller et al. (2016). The separation allows the investigation of the temporal evolution of the trace gas composition of the Ex-LS between the two phases. In the following, the water vapor and methane instruments aboard the HALO aircraft are briefly described. In addition, we used CLaMS (Chemical Lagrangian Model for the Stratosphere) (McKenna et al., 2002b) results interpolated onto the flight path, and also CLaMS backward-trajectory calculations from the measurement locations for identification of the air mass origin as described further below.

## 2.1 Water vapor measurements

The water vapor data used in this study were measured using the Fast In situ Stratospheric Hygrometer (FISH) instrument. The measurement technique is based on Lyman-$\alpha$ photofragment fluorescence and is suitable for measuring water vapor in the range of 1 ppmv to 1000 ppmv (Zöger et al., 1999). FISH is a well-established closed-path hygrometer with a long history in aircraft measurements and instrument intercomparisons (Meyer et al., 2015). Calibrations are performed during the campaigns with a MBW DP30 frost point hygrometer achieving accuracy of 6 % $\pm$ 0.4 ppmv at a time resolution of 1 Hz during TACTS/ESMVal.

## 2.2 Methane measurements

The methane data are obtained using the TRacer In situ QCL and HydrOgen Peroxide monitor (TRIHOP) instrument. The measurement technique is based on infrared absorption spectroscopy with a three channel quantum cascade laser spectrometer capable of measuring CO, $CO_2$, $N_2O$ and, $CH_4$. An in situ calibration against a secondary standard traceable to the NOAA scale was applied during the campaign. TRIHOP methane data are available every eight seconds with an integration time of 1.5 seconds, precision of 9.5 ppbv ($2\,\sigma$), and an uncertainty of 13.5 ppbv relative to the standard. For more details, see Müller et al. (2016).

## 2.3 CLaMS simulations with artificial air mass origin tracers

In this study, we used the three-dimensional chemistry transport model CLaMS (McKenna et al., 2002b, a; Pommrich et al., 2014, and references therein) according to the setup described in Vogel et al. (2015). The air parcels within the CLaMS model are represented by an ensemble of air parcels on a time-dependent irregular grid which are transported in a Lagragian way using trajectories including a physical representation of mixing (Konopka et al., 2004, 2010). This allows tracer gradients to be represented in a particularly accurate manner in CLaMS model simulations, making the model simulations well-suited for investigating transport and troposphere/stratosphere interaction (e.g., Konopka et al., 2010; Pommrich et al., 2014; Vogel et al., 2015, 2016).

The CLaMS simulation used here covers the time period from May 1 to October 31, 2012, and captures the Asian monsoon season in 2012. The simulation is driven by the meteorological fields of the European Reanalysis (ERA)-Interim (Dee et al., 2011) from the European Centre for Medium-Range Weather Forecasts (ECMWF) and covers the atmosphere from the surface up to 900 K potential temperature ($\approx$ 37 km altitude) with a vertical resolution of 400 m around the tropopause. In the simulation, "emission tracers" are released in the boundary layer ($\approx$ 2-3 km above surface) which globally marks all regions of the Earth's surface. These different artificial tracers are freshly emitted every 24 h in the boundary layer and their concentration in the atmosphere represents the air mass contributions from the respective source regions. For more details, see Vogel et al. (2015, 2016).

Vogel et al. (2015) showed that the emission tracers for India and eastern China are correlated with high values of potential vorticity (PV) and Aura MLS (Livesey et al., 2011) satellite measurements of $O_3$ and CO. Thus, they can be used as proxies for the location and shape of the AMA and allow transport studies of air masses from the AMA into the northern extra-tropical

**Table 1.** Latitude and longitude range of artificial boundary layer sources in the CLaMS model, which are used to define the "Asian monsoon surface tracer" (MON).

| Emission tracer | Latitude | Longitude |
|---|---|---|
| Northern India (NIN) | 20-40°N | 55-90°E |
| Southern India (SIN) | 0-20°N | 55-90°E |
| Eastern China (ECH) | 20-40°N | 90-125°E |
| Southeast Asia (SEA) | 12-20°N | 90-155°E |
| Tropical Pacific Ocean (TPO) | 20°S-20°N | 55°E-80°W |

lower stratosphere. In addition, the emission tracers from southeast Asia and the Pacific Ocean contribute to the air mass composition of the outer-edge of the AMA (Vogel et al., 2015, 2016). Thus, to study the transport of air masses affected by the Asian monsoon, the most important emission regions are India/China, southeast Asia, and the tropical Pacific Ocean. The sum of all these model boundary layer tracers is referred to as "Asian monsoon surface tracer" (MON) (see Table 1).

This tracer marks young air masses (younger than 5 months) that are confined during summer in the AMA and air masses from southeast Asia and the tropical Pacific Ocean influenced by the large-scale flow around the anticyclone in the UTLS. MON tracer preferentially accumulates with contribution up to 40% in and around the AMA and is clearly separated from the tropics and regions outside of the monsoon circulation with a contribution of around 10% (see Fig. 5 in Vogel et al., 2016). All other source regions can be neglected, since it was shown in Vogel et al. (2016) that these regions did not make a significant

contribution to the transport of young tropospheric air masses into the lower stratosphere in summer and autumn 2012.

## 2.4 Air mass trajectory calculations

In contrast to the three-dimensional CLaMS simulations, pure trajectory calculations do not account for mixing. However, the position of an air parcel and the changes in temperature along the trajectory can be tracked with sufficient accuracy over several days or even weeks (e.g., Vogel et al., 2014; Rolf et al., 2015). The air mass trajectories are calculated by means of the CLaMS

trajectory module for the last 50 days and from every 10th second from the location of the flight path using ERA-Interim data (Dee et al., 2011). The trajectories are based on the horizontal wind fields and diabatic heating rates with a resolution of 1° x 1° with 60 vertical levels from the surface up to 0.1 hPa. Additional parameters, such as temperature, pressure, and potential vorticity (PV) are interpolated from the ERA-Interim data onto the trajectories. In addition, the local distance to the thermal tropopause (WMO) is calculated for each trajectory time step and added to the trajectory data (e.g., Krebsbach et al., 2006). The

water vapor saturation mixing ratio with respect to ice (hereafter: saturation mixing ratio) along the trajectories is calculated based on the actual trajectory temperature. The Lagrangian cold point (LCP) is defined by the minimum of the temperature along the trajectory and thus the minimum saturation mixing ratio. With these parameters, pure stratospheric trajectories can be separated from trajectories with a tropospheric origin within the last 50 days by comparing the distance to the tropopause before and after the LCP. All trajectories with a negative distance to the thermal tropopause before the LCP and a negative

afterwards passed the tropopause from the troposphere into the stratosphere and represent air masses which originate in the troposphere.

Convection is the major processes to transport air masses vertically from the ground into the upper troposphere in the Asian monsoon region. In the model simulations shown here, vertical transport is taken into account as represented in ERA-Interim

including latent heat related transport (for pressure less than ≈300 hPa). Thus, the very small-scale rapid uplift in convective cores cannot included in CLaMS simulations and trajectory calculation. However, for the questions addressed here, which are concerned about large scale uplift in the monsoon, these small-scale features might be less relevant. In particular, previous studies using 3-D CLaMS simulations or trajectory calculations (e.g., Ploeger et al., 2015; Pommrich et al., 2014; Vogel et al., 2014, 2016; Müller et al., 2016; Ungermann et al., 2016), in comparison with satellite or in situ measurements, show that

ERA-Interim data are well suited for studying transport processes in the vicinity of the AMA and for CLaMS to reproduced every small scale feature in the measurements.

## 3 Results

The main purpose of this study is to corroborate the hypothesis that air masses from Asia and the tropical Pacific affected by the AMA are the main source of the moistening found in the Ex-LS in late summer and fall in the Northern Hemisphere, at

least for the campaign period. Firstly, we present in situ water vapor measurements of the two phases (August and September) and support the hypothesis based on water vapor - methane correlations. Secondly, we combine the measurements with CLaMS model results and the 50-day backward trajectory analysis to ultimately confirm the hypothesis.

### 3.1 Moistening the Ex-LS

Water vapor is strongly coupled to the temperature. The thermal tropopause in the sub-tropics and mid-latitude is found to

be a good indicator for marking the critical level of the temperature gradient and sharp relative humidity change, as shown by Pan and Munchak (2011), and thus separates the moist tropospheric regime from the dry stratospheric regime. In addition, isentropic transport is found to be the main pathway from the AMA into the Ex-LS (Vogel et al., 2016). Therefore, we choose the distance to the local thermal tropopause as the vertical coordinate in units of potential temperature. In Figures 2 a and 2 b, we display the mean water vapor distributions of both phases (phase 1 and phase 2) along the PV-based equivalent latitude

(eqLat). The use of EqLat allows us to combine observations performed at different times and locations by neglecting short-term dynamical variability (e.g. planetary wave activity at the sub-tropical jet). High water vapor mixing ratios (> 10 ppmv), representing moist tropospheric air, can be found below and up to 20 K above the thermal tropopause during both phases. Drier air masses down to the stratospheric background level of 4-5 ppmv are observed 30 K to 90 K above the thermal tropopause. In phase 1, the drier air masses are observed closer to the thermal tropopause, while in phase 2 such dry air masses are found

at potential temperatures 10 K higher. Both distributions have similar vertical and horizontal extents and thus the difference between both phases reveals local changes in water vapor mixing ratios over the time period (see Fig. 2c). Air masses with PV < 8 PVU reveal a large variability between phase 2 and phase 1, most likely due to the local variability of the thermal

tropopause not captured by the ECMWF data and planetary wave activity which distort the location of the sub-tropical jet core. Kunz et al. (2015) showed that the climatological transport barrier between tropospheric and stratospheric air masses is located at PV values of 7-8 PVU between June and November in the Northern Hemisphere. Thus, we will focus on air masses with PV > 8 PVU in the following analysis, since they are not affected by small-scale mixing processes directly at the tropopause.

All the air masses with PV values larger than 8 PVU show a distinct 5-10 % increase of water vapor in phase 2 compared to phase 1. The occurrence of these air masses ranges from 40°N to 75°N in equivalent latitude horizontally and corresponds to potential temperatures larger than 380 K. The frequency distributions of the water vapor mixing ratios with PV values larger than 8 PVU allow us to quantify the strength of moistening between both phases (see Fig. 2d). The frequency distribution of phase 1 is very compact with a mean water vapor mixing ratio of 4.5 ppmv, while the distribution of phase 2 is shifted to
higher water vapor mixing ratios with a mean value of 5 ppmv and shows a tail towards higher water vapor mixing ratios. The difference in the mean water vapor mixing ratio between the two phases indicates a statistically significant (Mann-Whitney-U-Test with p-value $< 10^{-100}$) moistening of the Ex-LS of about 0.5 ppmv within a time period of less than a month. This finding is in accordance with CLaMS model simulations and MLS satellite measurements which show a total increase of mean water vapor mixing ratio in the northern Ex-LS by 1.5 ppmv (1.0 ppmv) at 380 K (400 K) in the time period from May to November
due to the transport of air masses originating from source regions in Asia (see Fig. 15 in Vogel et al., 2016). The gradient of this increase was found to be of its strongest between the end of August and the end of September, which is the result of a pronounced eddy-shedding event on 20 September 2012 (Vogel et al., 2014, 2016). This strong increase perfectly matches the separation of the two phases and is in agreement with the increase observed in this study here. In addition to the measured increase in mean water vapor of 0.5 ppmv, a mean increase of 20 ppbv methane is observed in the same time period (see Fig.
20   4).

### 3.2    Tracer of air mass origin

Our simulations of corresponding artificial air mass origin tracers (Figures 3 a and 3 b) indicate that the moistening observed during phase 2 is the result of an increase in air mass contribution from the Asian monsoon region. MON increased by about 20 % to 100 % from phase 1 to phase 2 above the 8 PVU level (see Fig. 3 c). The increase of MON in this altitude range is
correlated with the observed increase in water vapor between both phases. The frequency distributions in Figure 3 d reveal the clear shift of MON to higher values, where the mean values show an increase of 8.6 % from phase 1 to phase 2. In particular, there is only a slight increase in surface tracers (~1 %) from other locations (residual surface without the monsoon region, MON) in the same time period. Therefore, the moistening of 0.5 ppmv can be linked to the 8.6 % increase of MON.

### 3.3    Water vapor - methane correlation

The correlation of observed water vapor with observed methane provides further evidence of the large air mass contribution of the Asian monsoon region. Methane has a source at the Earth's surface and thus has the highest concentrations in the troposphere. In the stratosphere, the methane mixing ratio decreases with altitude due to oxidization. Methane concentrations are especially high in regions with large industrial activity or natural out-gassing from wetlands, such as rice cultivation in

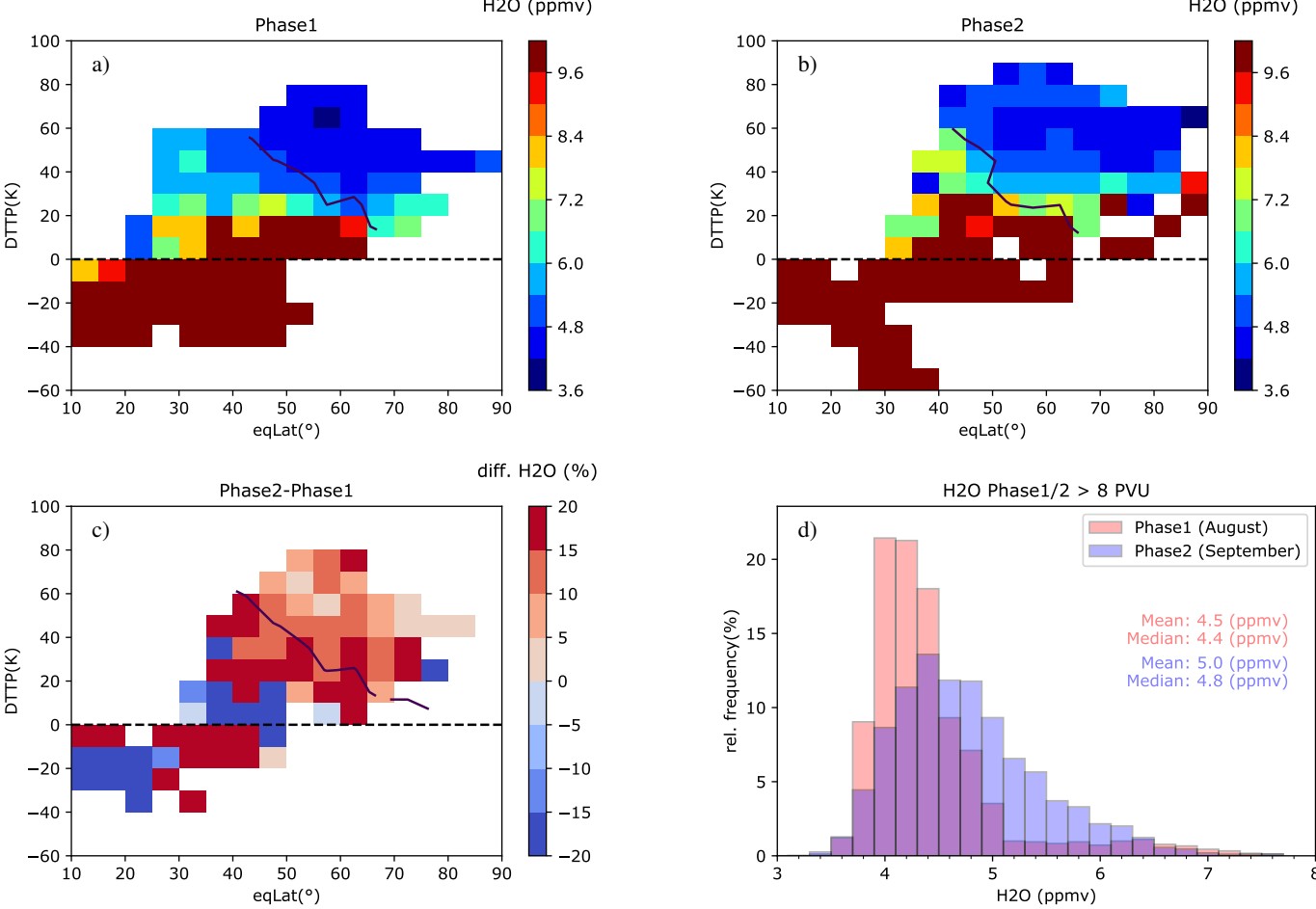

**Figure 2.** Mean distribution of H$_2$O as a function of equivalent latitude and distance to thermal tropopause (DTTP) in potential temperature units: a) phase and 1 b) phase 2. c) Relative differences of H$_2$O between phase 2 and phase 1. The dashed/solid black lines represent the thermal tropopause and the 8 PVU isoline, respectively. d) Frequency distribution of H$_2$O for the lower stratosphere (PV > 8 PVU) for phase 1 (red) and phase 2 (blue). The mean and median values of each distribution are given in red and blue, respectively.

India/Southeast Asia (Park et al., 2004; Xiong et al., 2009). Due to the high emissions on the ground, strongly enhanced mixing ratios of methane are found in the AMA compared to the free troposphere, as shown by the CARIBIC measurements (Schuck et al., 2010; Patra et al., 2011; Baker et al., 2012). Thus, methane is suitable as a monsoon-specific tropospheric tracer, in particular for the Asian monsoon regions. The difference in the methane distribution between both phases is visible in Figure 4. The frequency distributions look very similar to the ones found for water vapor and MON (Sec.3.1 and 3.2) and reveal a mean increase of 24 ppbv methane between both phases. This provides further evidence to combine water vapor and methane measurements together with the artificial origin tracer MON.

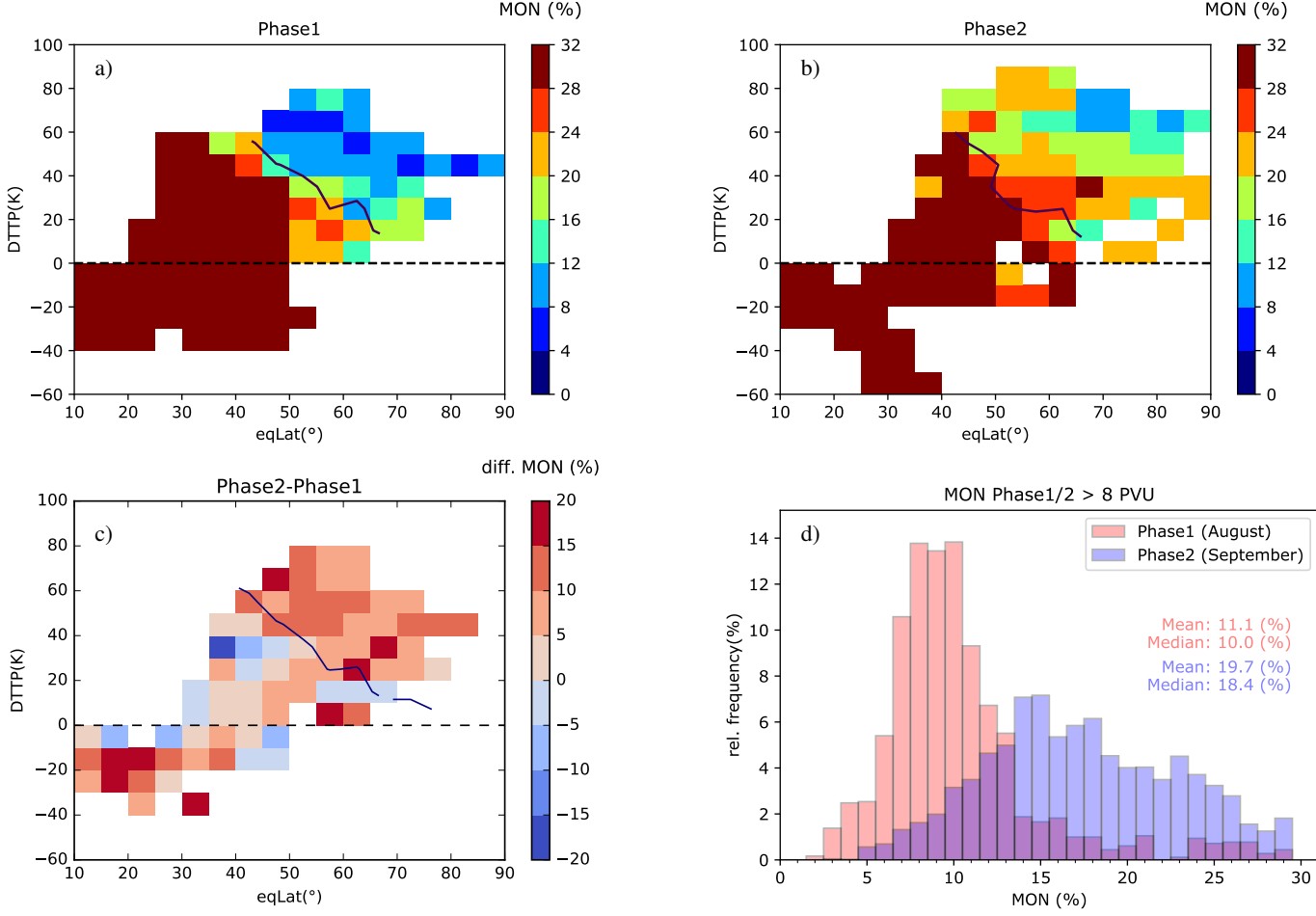

**Figure 3.** Same as Figure 2 but for the simulated Asian monsoon surface tracer MON.

Figure 5 shows the correlations of measured water vapor and methane mixing ratios in the lower stratosphere (PV > 8 PVU) for both phases. The core region (black contour) of the scatter plot encompasses 75 % of the data points of the underlying frequency distribution and ranges from around 3.8 ppmv to 5.1 ppmv in water vapor and from 1700 ppbv to 1730 ppbv in methane. The correlation in phase 1 is compact with nearly no significant enhancement in methane and water vapor compared to the water vapor background mixing ratio of 5 ppmv in the lower stratosphere (Zahn et al., 2014). The amount of MON is relatively low with values around 12 % in the core region. Only a few measurements (around 10% of the data) with high MON values of up to 30 % are found, showing the highest water vapor and methane values in phase 1. This indicates the moistening of the Ex-LS from the earlier phase of the monsoon before the end of August. In contrast, the correlation in phase 2 is dominated by measurements with higher methane and water vapor mixing ratios. The core region of the data ranges from 1700 ppbv to 1790 ppbv in methane and 3.7 ppmv to 6.2 ppmv in water vapor. The amount of MON in phase 2 is 30 % and thus significantly

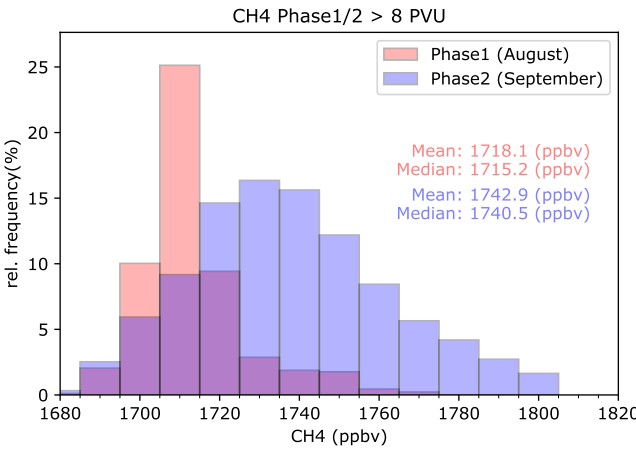

**Figure 4.** Frequency distribution of CH$_4$ for the lower stratosphere (PV > 8 PVU) for phase 1 (red) and phase 2 (blue). The mean and median values of each distribution are given in red and blue, respectively.

higher compared to phase 1. The water vapor and methane mixing ratio is statistically correlated with the MON tracer. In fact, the slope of the correlation is tilted towards higher methane mixing ratios in phase 2, which confirms the influence of methane-rich tropospheric air masses from the Asian monsoon region in the Ex-LS. The associated correlation coefficients for water vapor and methane are 0.79 and 0.67 for phase 1 and phase 2, respectively. The correlation coefficients for the measured water vapor and modeled MON tracer are 0.27 in phase 1 and 0.40 in phase 2, whilst for methane and MON tracer they are 0.41 for phase 1 and 0.63 for phase 2. All correlation coefficients (calculated according to Pearson) are highly significant with p-values of nearly zero ($< 10^{-18}$ ), thus rejecting the null hypothesis of an uncorrelated dataset. In fact, the correlation for water vapor and methane is clearly higher compared to the correlation with the MON tracer, which corroborates the consistency between both in situ measurements. It also shows the limitations of CLaMS to reproduce every small-scale feature in the measurements. However, the correlation for the model-based tracer and the measurements is still quite satisfactory and strongly supports the hypothesis that the increase in water vapor is correlated to the air mass affected by the AMA.

In addition, methane shows a stronger correlation to MON due to its long life time of around 9 years in the troposphere and even longer life time in the lower stratosphere (Wuebbles and Hayhoe, 2002). It also reacts similarly to the ideal model tracer MON. In contrast, water vapor has a very short life time of about 9 days (Hartmann, 2015) due to its strong temperature dependency. Therefore, we can expect a reduced variability of water vapor in comparison to methane and thus a better correlation of methane with the MON tracer. This provides the motivation for showing the connection of measured water vapor mixing ratios to their air mass temperature history in the next section.

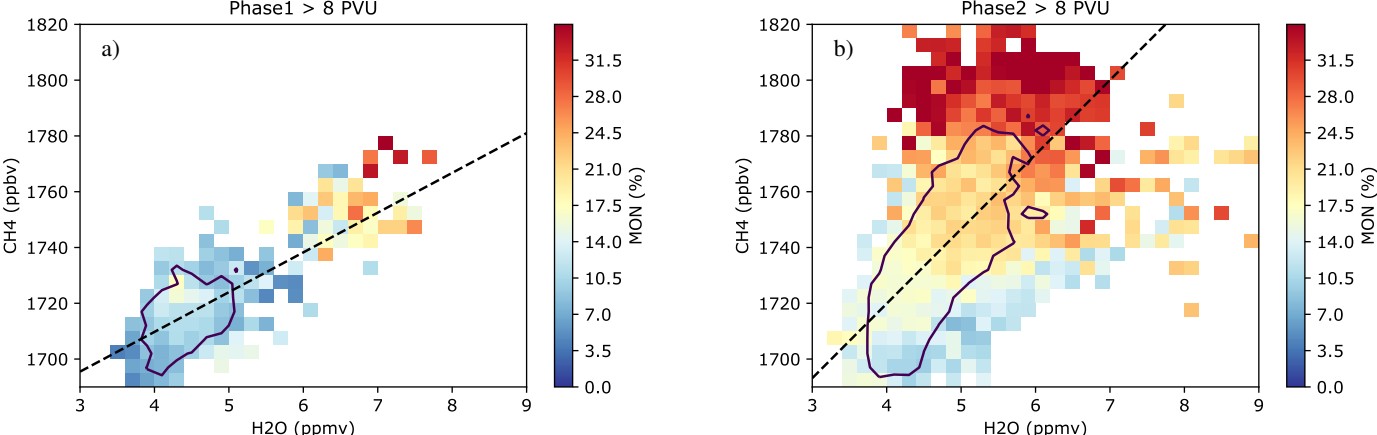

**Figure 5.** Correlation of water vapor and methane for PV > 8 PVU, color-coded by the Asian monsoon surface tracer MON. The solid black line marks the core region, which encompasses 75 % of the data. The black dashed line represents a robust linear regression fitting curve (Theil–Sen regression).

## 3.4 Lagrangian cold point from backward trajectories

The amount of water vapor transported from the troposphere into the stratosphere is strongly coupled with the Lagrangian cold point (LCP), where typically the air masses are dehydrated close to the saturation mixing ratio by ice crystal formation and subsequent sedimentation (e.g., Schoeberl and Dessler, 2011). Thus, the amount of water vapor in these air masses is nearly
5   preserved after passing the LCP in the tropical, sub-tropical, and mid-latitude stratosphere. The geographic location of the LCP along the backward trajectory marks the location of these imprints on the water vapor mixing ratio.

Vogel et al. (2016) showed that the eddy shedding and in-mixing of tropospheric air masses can occur at potential temperatures of around 380 K directly between double tropopause features, passing the transport barrier at the climatological jet core. However, the locations of LCPs in the case of eddy shedding events do not implicitly coincide with the point of in-mixing
10   into the Ex-LS (Schoeberl and Dessler, 2011; Vogel et al., 2016). In this case, the locations of LCPs are typically at the top of the upwelling in the AMA region, whereas in-mixing into the Ex-LS can be found at locations where Rossby wave breaking occurs eastwards or westwards of the AMA along the subtropical jet.

In the following, all backward trajectories for air parcels with PV > 8 PVU according to Section 3.1 are considered (i.e., 2909 out of 11333 (25.6 %) trajectories for phase 1 and 7301 out of 14673 (49.8 %) trajectories for phase 2). The LCPs (see
15   Figure 6 a and 6 b) are mostly located in the subtropics with equivalent latitudes ranging from 0°N to 50°N. They cluster meridionally above India/China/Indonesia, North America, and West/North Africa in both phases. Interestingly, all these regions can be assigned to respective monsoon systems over Asia, America, and Africa. In phase 1, only 158 out of 2909 (5.4 %) trajectories with PV > 8 PVU are classified as tropospheric with the LCP criteria above, i.e., more trajectories are entirely in the stratosphere. Furthermore, phase 2 reveals more frequent LCP occurrence with 846 out of 7301 (11.6 %) trajectories with

PV > 8 PVU in these regions. In addition, phase 2 also shows a clear increase of LCPs in the region of India/China/Indonesia compared to phase 1. This is further confirmed by the relative difference between phase 2 and phase 1, normalized to all trajectories (tropospheric and stratospheric, see Figure 6c). The signal of the two other monsoon circulations, i.e., the American monsoon and the African Monsoon, disappears in this analysis. The dominant appearance of LCPs in the Asian monsoon region is evident. Therefore, the Asian monsoon has the most frequent LCP occurrence, the largest extent, and thus the largest impact on stratospheric water vapor enhancements in the observed time period.

The selection criterion for trajectories described in Section 2.4 can be used to quantify the amount of trajectories which remained in the stratosphere for at least 50 days (stratospheric origin) or passed the LCP from the troposphere into the stratosphere (tropospheric origin), as shown in Figure 7. In phase 1, the fraction of flight path back-trajectories with tropospheric origin ($p1t$) is 5.4 % (94.6 % stratospheric, $p1s$), while in phase 2, 11.6 % ($p2t$) had a tropospheric origin (88.4 % stratospheric, $p2s$). The larger fraction of tropospheric origin trajectories corroborate the existence of stronger troposphere to stratosphere exchange contributing to phase 2. The increase in mean water vapor from phase 1 (4.5 ppmv) to phase 2 (5.0 ppmv) by the Asian monsoon trajectories can be calculated with a system of linear equations:

$$\begin{pmatrix} p1t & p1s \\ p2t & p2s \end{pmatrix} \begin{pmatrix} wv_1 \\ wv_2 \end{pmatrix} = \begin{pmatrix} 4.5\,\text{ppmv} \\ 5.0\,\text{ppmv} \end{pmatrix} \quad \Rightarrow \quad wv_1 = 4.1\,\text{ppmv}, wv_2 = 11.9\,\text{ppmv} \tag{1}$$

Here, $wv_1$ denotes the water vapor background mixing ratio and $wv_2$ the water vapor contribution from the monsoon trajectories. From this simple calculation, it can be concluded that the observed increase of 0.5 ppmv in the mean water vapor mixing ratio between both phases (see Figure 2d) can be attributed to the transport of moist tropospheric air masses with a mean water vapor mixing ratio of about 12 ppmv compared to a stratospheric background of 4.1 ppmv. These values fit quite well with the CLaMS model simulations in Vogel et al. (2016), who determined a stratospheric background of around 4 ppmv in the same time period neglecting water vapor contributions from air masses in the Asian monsoon region.

## 4    Conclusions

The mean water vapor/methane increase of 0.5 ppmv (11 %)/20 ppbv (1.2 %) from phase 1 (August) to phase 2 (September) is correlated with an increase in Asian monsoon surface tracer (MON) by 8.6 percentage points, which corresponds to a doubling of air masses in the Ex-LS influenced by the region of the Asian monsoon, southeast Asia, and tropical Pacific Ocean. Other surface regions can be neglected as a source of the moist air masses in the stratosphere during Northern summer. The doubling of MON is also consistent with observations of increased values of other tropospheric tracers ($N_2O$, $SF_6$, CO), which were simultaneously observed (Müller et al., 2016). Our observational results also confirm the modeling study of Ploeger et al. (2013), where a simulated increase of the water vapor from 4.5 ppmv to 5.0 ppmv in the Ex-LS corresponds to an increase of the Asian monsoon air contribution of about 10 %. Our observed increase in water vapor is also in accordance with the study by Vogel et al. (2016), who found that over the entire monsoon time period from June to October 2012, 1.5 ppmv/1.0 ppmv of the water vapor in the northern hemispheric stratosphere at 380 K/400 K originates from the AMA regions including southeast

Asia and the tropical Pacific Ocean. The gradient of this increase was found to be at its strongest between the end of August and the end of September, similarly to the two phases used in this study.

Moreover, the flight path back-trajectories support the influence of the Asian monsoon on the Ex-LS. In phase 2, twice as many trajectories show a tropospheric influence compared to phase 1. The moistening in the Ex-LS is maintained by young air masses transported from the Asian monsoon with a mean mixing ratio of about 12 ppmv. The locations of the LCPs show that the imprint of in-mixed water vapor mixing ratios occurs most frequently above India and south-east Asia. The crossings of the tropopause in the case of eddy-shedding events are displaced to Rossby wave breaking regions along the subtropical jet. From these points on, air masses are transported quasi-isentropicaly into the Ex-LS as shown by Vogel et al. (2016).

All these measurements and model results contribute to a general understanding that the Asian monsoon influences the trace gas composition of the extra-tropical lower stratosphere in the Northern Hemisphere – at least for the year 2012. In particular, this study provides observational evidence of the water vapor transport from the Asian monsoon region to the northern Ex-LS. Following the study by Riese et al. (2012), a 5-10% increase of water vapor in the Ex-LS produced a top of the atmosphere radiative forcing of 100-500 mW m$^{-2}$. Depending on the residence time in the lower stratosphere, the moist and methane-rich air masses have the potential to impact surface temperatures in the Northern Hemisphere.

*Acknowledgements.* We would like to thank the coordinators of the two HALO missions TACTS and ESMVal, namely Andreas Engel, Harald Bönisch (University of Frankfurt), and Hans Schlager (DLR), for their efforts. Special thanks go to DLR-FX for the avionic data from the Bahamas instrument, Heini Wernli for providing tropopause information, and Jens-Uwe Grooß for CLaMS model forecasts and flight planning within the LASSO project (HALO-SPP,1294/GR,3786) supported by the German Research Foundation (DFG). We would also like to thank the Jülich Supercomputing Centre (JSC) for computing time on the supercomputer JUROPA within the VSR project JICG11. Our activities were part of the DFG's AMOS project (HALO-SPP 1294/VO 1276/5-1) and the Seventh Framework Program (FP7/2007–2013) of the European Union's StratoClim project (grant agreement no. 603557). Thanks also go to the Language Service of Forschungszentrum Jülich for linguistic revision. In addition, we gratefully acknowledge the ECMWF for their meteorological reanalysis data support. We thank the two anonymous reviewers for their careful reading of our manuscript and their helpful comments and suggestions. The observational data used in this study can be downloaded from the HALO database (DOI:10.17616/R39Q0T) at https://halo-db.pa.op.dlr.de/.

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

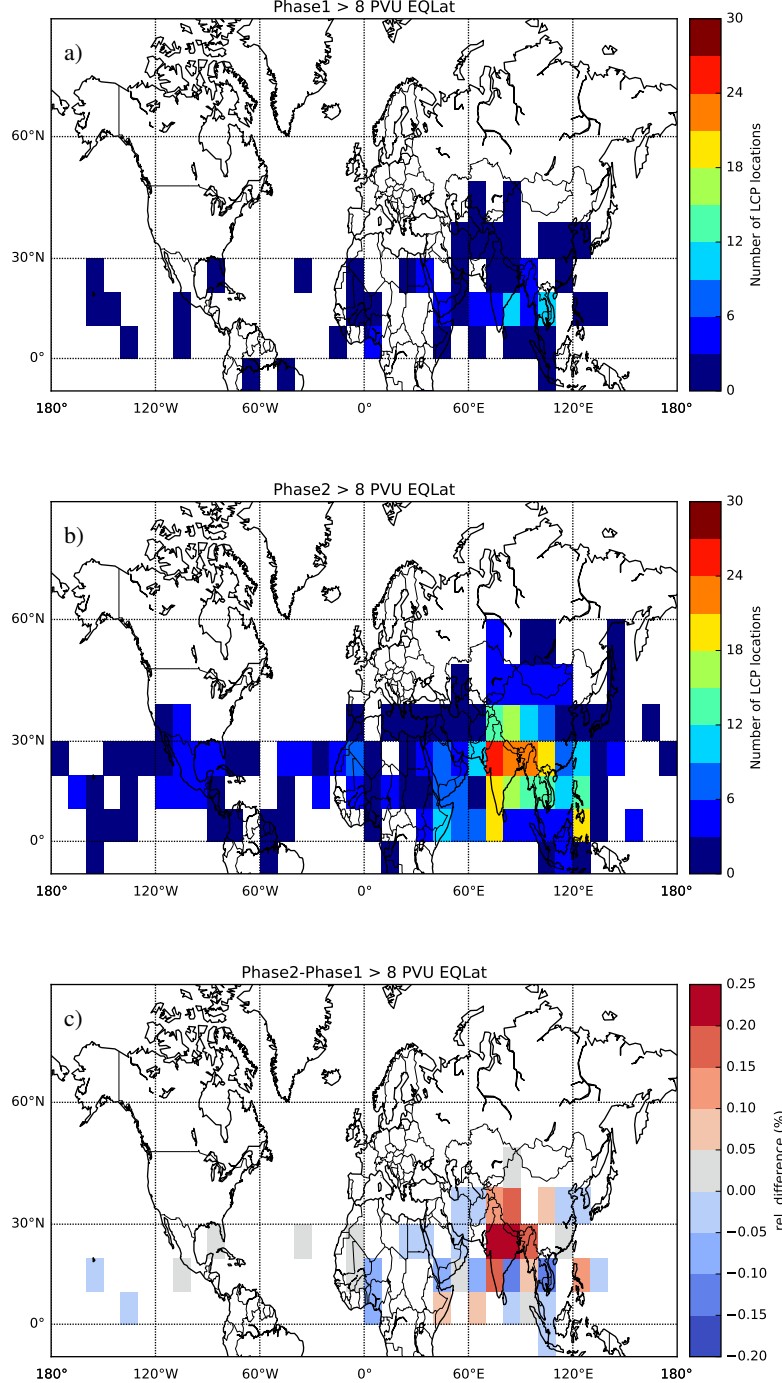

**Figure 6.** Location of Lagrangian Cold point (LCP) binned by longitude and latitude. LCP locations for phase 1 and phase 2 are shown in a) and b), respectively. The relative difference normalized to the total number of trajectories is shown in Fig. c).

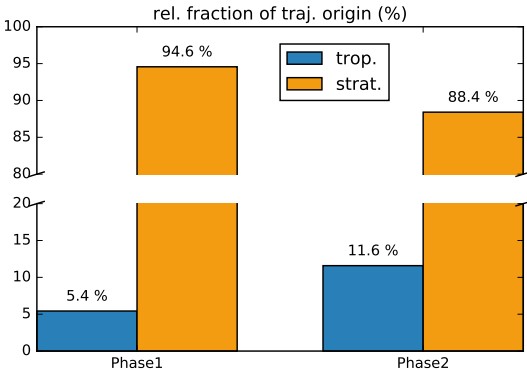

**Figure 7.** Fraction of trajectories with tropospheric (blue) or stratospheric (orange) origins for each phase.