# Peer review of "Water vapor increase in the lower stratosphere of the northern hemisphere due to the Asian monsoon anticyclone observed during the TACTS/ESMVal campaigns"

_Atmospheric Chemistry and Physics, 2017_

## Referee Comment (RC1) · Anonymous Referee #1 · 13 Nov 2017

General Comments

This study presents results of the comprehensive analyses on water vapor transport in the upper troposphere and lower stratosphere related to the Asian summer monsoon anticyclone. Quantitative analyses are done by utilizing both the in-situ measurements during the TACTS/ESMval field campaigns and a Lagrangian chemistry transport model, CLaMS, during the summer of 2012. Although the results are presented in a reasonable manner, I would recommend refining some of the scientific questions and especially improving writing before it is considered to be publishable on the ACP. Below are my suggestions.

[Figure]

- I think it is necessary to include more background in the introduction. This may include, but not limited to, characteristics of the monsoon anticyclone and why the transport in and near the Asian monsoon anticyclone is important in global scale circulation.

- It is not clear what the scientific goals are. Why is the quantitative analysis of changes in the water vapor amount in the extra-tropical lower stratosphere important? Is 10

- Also, explain why in-situ measurements of chemical species are necessary and unique in this study compared to the previous studies. What are the limitations of using satellite measurements of water vapor in this type of study?

- Explain why water vapor and methane are used and what we can learn from those. What are the lifetimes of water vapor and methane in the troposphere and stratosphere? What other species were measured during the field campaign?

- Why is CLaMS used here? What is unique about this model? And are there any modeling studies on the similar subject?

- Writing can be improved. Careful selection of words should help clarifying some of the ideas. I think simple grammatical errors can easily be avoided by putting more effort into proof reading.

Specific Comments

- P1, L1 – from Asia -> locally

- P1, L2 – water vapor (H2O) of about 0.5 ppmv (11

- P1, L4 – 'by in-situ instrumentation in the northern hemisphere' is redundant

- P1, L4 – What are the full names of TACTS and ESMVal?

- P1, L5 – this water vapor and methane increase -> the increased water vapor and methane, with the help of -> using

- P1, L6 – transport model (CLaMs)

- P1, L9 – influence of air. . .region. -> influence due to the Asian monsoon anticyclone.

- P1, L10 – remove 'between'

- P1, L15 – gases like water vapor and methane -> gases, such as, water vapor and methane,

- P1, L19 – with a low -> with low

- P2, L3 – 'higher values. . .temperature' – The meaning of this is not clear.

- P2, L10 – by a potential -> by strong potential

- P2, L15 (and others) – Eddy shedding -> eddy shedding

- P2, L18 – AURA-MLS -> the Aura Microwave Limb Sounder (MLS)

- P2, L22 – . . .altitude range -> add a few references here.

- P2, L27 – This study bases on -> This study is based on the (or In this study, the data collected. . . are analyzed. . .)

- P2, L27 – Also what other species were measured during TACTS and ESMVal?

- P2, L28 (and others) – northern hemisphere -> Northern Hemisphere

- P2, L33 – A reference for CLaMS is necessary here. Also, why is CLaMS perfect for this type of research?

- P3, L1 – location -> locations

- P3, L5 – bases -> is based

- P3, L11 – The full name for TRIHOP should be given here. Also bases -> is based

- P3, L20 – is driven by ERA-Interim -> is driven by the European Reanalysis (ERA)-Interim

- P3, L22 – What are the definitions of "emission tracers"?

- P3, L24 – concentrations -> concentration

- P3, L27 – I do not think PV is measured by the MLS.

- P3, L27 – proxy -> proxies, allow for transport -> allow transport

- P3, L29-30 – Either add references here or explain how this can be done.

- P3, L30 – regions are -> regions considered are

- P4, Table 1 – A map can be added here. Or this table can be replaced by a map.

- P4, L3-4 – that these…stratosphere -> that these regions do not make a significant contribution to the lower stratosphere

- P4, L7-8 – In contract to three -> In contrast to the three, temperature development -> changes in the temperature

- P4, L8 – A reference is needed at the end of this sentence.

- P4, L10 – remove 'of the ECMWF'

- P4, L22 – 'the tropical Pacific' – I am not sure if this has been mentioned previously.

- P5, L1 – 'reinforce the hypotheses' – What does this mean?

- P5, L6 – Figures 1a and b -> Figures 1a and 1b (also P9, L8)

- P5, L7 – along equivalent latitude -> along the equivalent latitude. Also explain how 'equivalent latitude' is defined.

- P5, L8 – What does 'tropospheric influenced air' mean? Does this mean the air is mixed with tropospheric air?

- P5, L11 – have a similar…extent… -> have similar vertical and horizontal extents

- P5, L16 – at a PV value -> at PV values, remove 'in the time'

- P5, L19 – All air -> All the air

- P5, L22 – The distribution -> The frequency distribution

- P5, L25 – vapor concentration -> vapor concentration between the two phases

- P5, L28 – It might be easier to refer a figure in Vogel at al., instead of a page number.

- P5, L32 – Figures 2a, b -> Figures 2a and 2b

- P5, L33 – remove 'relatively'

- P6, Figure 1 – a)-d) can be written on the top left instead of bottom left (also in other figures).

- P6, L3 – Figure 2d) -> Figure 2d

- P6, L4 – from phase 2 to phase 1 -> from phase 1 to phase 2

- P8, L2 – enhancements -> enhancement

- P8, L5 – Only few -> Only a few

- P8, L7 – Explain what 'core region' means.

- P8, L10 – 'the slope of' – A linear regression line can be added in Fig. 3a.

- P8, L18 – What does 'the imprint on water vapor' mean here?

- P9, L1 – that Eddy shedding -> that the eddy shedding

- P9, L5 – A reference can be added at the end of this sentence.

- P10, L2 – between both phases -> from phase 1 (Aug) to phase 2 (Sep)

- P10, L7 – model study of -> modeling study of

- P10, L20 – gives observational -> gives an observational

---

## Referee Comment (RC2) · Anonymous Referee #2 · 15 Nov 2017

**General comments**

The paper of Rolf et al. presents analyses of aircraft in-situ measurements of water vapor and methane in the northern extra-tropical lower stratosphere (Ex-LS). Trends of $H_2O$ and $CH_4$ within the observational period are attributed to tropospheric inmixing in the Asian monsoon anticyclone (AMA) and subsequent transport to the region of the measurements, using the CLaMS model.

The topic of the paper is relevant and well within the scope of ACP(D). The current manuscript is well structured, but in places the writing impedes understanding of what the authors want to convey. So if one of the following comments seems to miss the

point completely, it might just be a language issue.

My main concern is the novelty or substance of the paper. Moistening of the Ex-LS via the AMA has been reported before (Randel and Jensen, 2013), even based on in-situ data from the same 2012 campaign and a similar methodology (Müller et al., 2016, Vogel et al., 2016). Vogel et al. (2016) also already quantified the moistening for the entire monsoon season of 2012. Although Rolf et al. provide additional and detailed analyses, new findings should be brought out better in the paper.

The analyses of this paper are based on measurements in a rather limited region and during a short period of time. It's a case study, which is not always properly reflected by the writing. Additional modelling (or observations) might help to scale up the results and draw more general conclusions, e.g. if the results hold for several monsoon seasons; or provide an estimate for the impact of the Ex-LS moistening on surface temperatures ... just ideas for adding some substance.

*Methodology:*

(1) I don't really see the point in using tracers of airmass origin here. The tracer(s) described in the paper do not seem to be reliable indicators of AMA air, and do not provide additional relevant information as compared to back-trajectories.

(2) The choice of the two time periods for detecting changes to H2O and CH4 seems to be rather arbitrary. The delta should be discussed in the context of the transport time scales involved, which might be available from back trajectories. Transport time scales also seem to be the key for relating the results of this paper to the numbers found by Vogel et al. (2016).

(3) Convection is an important process when it comes to the AMA, but is notoriously hard to capture in large-scale models. Please discuss, how you consider convection.

(4) The term "statistic" seems to have been added to the corresponding terms in re-

sponse to the initial review, but I still do not see actual statistical calculations. Please provide details on the statistical methods used for those obviously non-Gaussian distributions, mark significant points/ranges where applicable, provide correlation coefficients, confidence intervals etc. This might not apply to each and every use of statistical terms (correlation, significance, …), but you should make clear where it is normal language rather than backed by maths.

(5) Flight paths determine the region of study and should be shown. Citing Müller et al. (2016) only helps partially, because they analyse slightly different periods.

(6) Please distinguish between "concentration" and "mixing ratio" throughout the paper. For instance, "ppbv" is a unit for mixing ratios (better: nmol/mol). The SI unit for molar concentration is mol/m3, for number concentration 1/m3.

**Specific comments**

P1L8: Are those the exact numbers? Otherwise please use "about"

P1L13: exclusively -> mostly/predominantly

P2L9: AMA is leaky

P2L29: ESMVal went around Africa

P2L30: Why those dates? The Monsoon starts in June/July, but your approach seems to be based on the assumption that less trajectories from the AMA have reached the flight paths at the beginning of Sep compared to the end of Sep. Please discuss the time scales for transport from the AMA to the measurements.

P3L12: explain acronym

P3L28: Does that mean that this type of emission tracer preferentially ends up in the AMA? For using MON as a proxy for the AMA, it is not enough to show that MON is the main source of AMA air. Additionally, MON must not end up in significant amounts

outside the AMA.

P4L17: Does possible supersaturation play a role for these analyses?

P4L20: Consider rewording to make clear that both conditions must be met.

P5L9: The motivation for choosing potential temperature difference to the local thermal tropopause is not clear to me. Water vapour depends on temperature, but the thermal tropopause is based on a lapse rate. Also, potential temperature is good for characterizing isentropic transport, but not necessarily temperature history.

P5L10: Please briefly motivate using equivalent latitude

P5L11: consider choosing the contours in Fig. 1 to display the threshold used in the text

P5L15: Fig. 1 only shows equivalent latitude as vertical coordinate. Were similar longitudinal regions sampled during both phases?

P5L16: This sentence is not clear to me. Do you mean that below 8PVU some "pixels" in your eqLat-DTTP-space show larger differences than others of H2O between the two phases? How does the thermal tropopause affect local variability? Fig. 1c shows local differences in water vapour. It sounds a bit odd to attribute local differences of water vapour to local variability of water vapour. Consider revising.

P5L26: You point out that the distribution is compact for period 1. This gives the impression that most of the moistening happens between the two periods. However, period2 is already at the end of the monsoon season. Why is 0.5ppmv consistent with 1-1.5ppmv then? What are the transport time scales involved?

P5L28: How did you test significance for those non-Gaussian distributions?

P5L28: Are your measurements representative of the entire Ex-LS?

P5L33: Please show CH4. It might support your claim that the signature originates in

the AMA.

P6L4: 100

P6L5: Do you just want to say increasing H2O corresponds to increasing MON? However, providing some correlation coefficients for the region above 8PVU would be good. To me the pattern above 8PVU looks different between Figs. 1c and 2c.

P7L8: "Thus" and "because" in one sentence is confusing. Please clearly express why you consider CH4 to be a monsoon tracer. Please provide references for each argument.

P8L4: What is the statistically significant region in this plot? Please provide numbers for the background mixing ratios.

P8L5: The grammar of this sentence is odd. However, no matter how I interpret it, the definition of the core region remains unclear. If the core region is defined by the 75

P8L11: sounds odd, consider rewording

P8L12: This sentence is not clear to me. Did you test statistical significance? If so: How exactly did you do that? The scatter plots show H2O mixing ratio versus CH4 mixing ratio. If you want to discuss the statistics of deltas ("increase"), please show deltas.

P8L12: slope CH4/H2O -> tilted towards a higher ratio or higher CH4, but not towards higher H2O. Please revise. Also, you discussed that contributions from the ASM increase H2O and CH4. Fig. 3 looks like the ratio CH4/H2O changes. Is that also a tell-tale sign of ASM origin?

P8L17: 1. "Water vapor is dehydrated" sounds odd; 2. Do you mean that dehydration typically happens close to the saturation mixing ratio, or that air masses typically get close to saturation at the LCP? (Only wet air might produce ice crystals). Please revise.

P9L2: Are you considering only trajectories that originate in the troposphere here? The

LCP that determined water vapor might lie further back than the reach of the backward trajectory.

P9L10: remove "notably" or "interestingly"

P9L20: ... and region of the flights?

P9L23: Fig. 5 can be removed. The next sentence contains all the information.

P9L25: amount -> fraction or percentage

P9L25: in phase -> contributing to phase (Please have in mind that transport from the AMA needs some time.)

P10L5: How did you calculate the correlation? Please provide numbers.

P10L6: distinguish between AMA and the respective surface tracers

P10L11: I am not convinced. Please elaborate on how you relate your results to Vogel et al. (2016). Is water vapour transport from the AMA to the ex-LS constant throughout the monsoon season? How do you relate your phase1/gap/phase2 periods to the monsoon period? Are the diagnostics comparable?

P10L19: crossing is displaced OR crossings are displaced; What is combined with quasi-isentropic transport? Please revise and consider splitting it into two sentences.

P10L21: "picture of influencing" sounds odd

P10L25: This should be shown, at least via a rough estimate.

**References:**

Müller, S., P. Hoor, H. Bozem, E. Gute, B. Vogel, A. Zahn, H. Bönisch, T. Keber, M. Krämer, C. Rolf, M. Riese, H. Schlager, and A. Engel (2016), Impact of the Asian monsoon on the extratropical lower stratosphere: trace gas observations during TACTS over Europe 2012, Atmospheric Chemistry and Physics, 16(16), 10,573–

10,589, https://doi.org/10.5194/acp-16-10573-2016.

Randel, W. J. and Jensen, E. J.: Physical processes in the tropical tropopause layer and their roles in a changing climate, Nat. Geosci, 6, 169–176, doi:10.1038/ngeo1733, 2013.

Vogel, B., G. Günther, R. Müller, J. U. Grooß, A. Afchine, H. Bozem, P. Hoor, M. Krämer, S. Müller, M. Riese, C. Rolf, N. Spelten, G. P. Stiller, J. Ungermann, and A. Zahn (2016), Long-range transport pathways of tropospheric source gases originating in Asia into the northern lower stratosphere during the Asian monsoon season 2012, Atmospheric Chemistry and Physics, 16(23), 15,301–15,325, https://doi.org/10.5194/acp-16-15301-2016.

---

## Author Comment (AC1) · 9 Jan 2018

The authors thank reviewer #1 for the many fruitful comments which were helpful to improve the paper. All changes to the paper are highlighted in red color. Point by point answers to your comments are reported below.

**Major comments**

- *- I think it is necessary to include more background in the introduction. This may include, but not limited to, characteristics of the monsoon anticyclone and why*

*the transport in and near the Asian monsoon anticyclone is important in global scale circulation.*

We included more information about the Asian monsoon and to transport pathways to better introduce the background and to motivate this study. For example, we now discuss the papers by Randel and Jensen, 2013, Ungermann et al. 2016.

- *- It is not clear what the scientific goals are. Why is the quantitative analysis of changes in the water vapor amount in the extra-tropical lower stratosphere important? Is 10*

 We added more information about water vapor and motivate why the knowledge about small changes of water vapor concentration in the UTLS are important for global climate like the study by Solomon et al. 2010.

- *- Also, explain why in-situ measurements of chemical species are necessary and unique in this study compared to the previous studies. What are the limitations of using satellite measurements of water vapor in this type of study?*

 We already explained that especially satellites, like limb sounders, have a coarse vertical resolution. We added a sentence to the introduction to stronger motivate the use of in-situ instrumentation: Especially in the tropospause region this coarse resolution can smooth the strong vertical gradient of water vapor at the tropopause and may lead to both an over- or underestimation of the water vapor concentration in the lower stratosphere. Here, we present high resolution and precise in-situ water vapor and methane measurements in the northern lower stratosphere performed in August and September 2012.

- *- Explain why water vapor and methane are used and what we can learn from those. What are the lifetimes of water vapor and methane in the troposphere and strato- sphere? What other species were measured during the field campaign?*

 First, water vapor is a major green house gas and the variablity of water vapor at tropopause altitudes has a strong impact on surface climate (Solomon et al.

2010, Riese et al. 2012). This is a major motivation of our study which is now included in the introduction. Second, Methane has a long lifetime of around 8.9 years in the troposphere and even longer in the lower stratosphere (Wuebbles et al. 2002), whereas water vapor has a very short life time of about 9 days (Hartmann 2015) due to its strong temperature dependence. Therefore water vapor is very variable with respect to transport history, in contrast to methane. This is also the reason why methane shows a stronger correlation to the MON tracer compared to water vapor. Both correlation coefficients are now included in the text. In addition, both trace gases are important green house gases and small changes in their concentration in the UTLS have strong climatic relevance. We included now the life times and more motivation concerning the choice of trace gases in the introduction and section 3.3.

- *- Why is CLaMS used here? What is unique about this model? And are there any modeling studies on the similar subject?*
  For the scientific issues addressed here, a model is required that provides a good resolution in the UTLS and, in particular, describes well quasi-isentropic transport in the UTLS as well as transport in the presence of strong tracer gradients. Due to its Lagragian transport on isentropic surfaces with a physically based mixing scheme (that avoids the classic problem of numerical diffusion in Eulerian transport schemes) CLaMS is indeed very well suited for the scientific questions addressed in our paper. CLaMS is indeed widely used like Vogel et al., ACP, 2015, Vogel et al., ACP, 2016, Ploeger et al., JGR, 2013, Pommrich, GMD, 2011, Pommrich, GMD , 2014, Vogel et al., JGR, 2012, Konopka et al., ACP, 2010, Hoppe et al., GMD, 2014. Some (not all) of these studies are now cited in the text.

- *- Writing can be improved. Careful selection of words should help clarifying some of the ideas. I think simple grammatical errors can easily be avoided by putting more effort into proof reading.*

We agree with the reviewer and thank her/him for the many good and helpful suggestions. We carefully checked the writing and sent this manuscript to the language office at our institution for proof reading by a native speaker.

**Specific comments**

- *- P1, L1 – from Asia -> locally*
  Changed.

- *- P1, L2 – water vapor (H2O) of about 0.5 ppmv (11*
  Changed.

- *- P1, L4 – 'by in-situ instrumentation in the northern hemisphere' is redundant*
  Changed.

- *- P1, L4 – What are the full names of TACTS and ESMVal?*
  We added the full names in the text of Section 2. We think in the abstract, is not the right place resolve acronyms of minor importance.

- *- P1, L5 – this water vapor and methane increase -> the increased water vapor and methane, with the help of -> using*
  Changed.

- *- P1, L6 – transport model (CLaMS)*
  Changed.

- *- P1, L9 – influence of air. . .region. -> influence due to the Asian monsoon anticyclone.*
  Changed.

- *- P1, L10 – remove 'between'*
  Changed.

Interactive
comment
- *- P1, L15 – gases like water vapor and methane -> gases, such as, water vapor and methane,*
  Changed.

- *- P1, L19 – with a low -> with low*
  Changed.

- *- P2, L3 – 'higher values. . .temperature' – The meaning of this is not clear.*
  Changed.

- *- P2, L10 – by a potential -> by strong potential*
  We did not add the word strong, because the transport barrier is often rather leaky as shown in Ploeger et al. 2015 and the gradient is much lower than at the polar vortex for example.

- *- P2, L15 (and others) – Eddy shedding -> eddy shedding*
  Changed.

- *- P2, L18 – AURA-MLS -> the Aura Microwave Limb Sounder (MLS)*
  Changed.

- *- P2, L22 – . . .altitude range -> add a few references here.*
  Changed.

- *- P2, L27 – This study bases on -> This study is based on the (or In this study, the data collected. . . are analyzed. . .)*
  Changed.

- *- P2, L27 – Also what other species were measured during TACTS and ESMVal?*
  There were other species like CO, N2O, and O3 measured during TACTS and ESMVal. The results are published in Müller et al. 2016. In this study, we particularly highlight the importance of the water vapor transport associated to the AMA.

- *- P2, L28 (and others) – northern hemisphere -> Northern Hemisphere*
Changed.

- *- P2, L33 – A reference for CLaMS is necessary here. Also, why is CLaMS perfect for this type of research?*
We included a reference in the text and put more information as well as a brief motivation in the sub section about CLaMS. See also the answer to your general comment #5.

- *- P3, L1 – location -> locations*
Changed.

- *- P3, L5 – bases -> is based*
Changed.

- *- P3, L11 – The full name for TRIHOP should be given here. Also bases -> is based*
Both is changed.

- *- P3, L20 – is driven by ERA-Interim -> is driven by the European Reanalysis (ERA)- Interim*
Changed.

- *- P3, L22 – What are the definitions of "emission tracers"?*
The "emission tracers" are artificial tracers with no chemical depletion, but are transported within the atmosphere. During this transport mixing is applied, thus these tracers can be diluted by emission tracers from different source regions. More information can be found in the cited literature of Vogel et al. 2015 and 2016.

- *- P3, L24 – concentrations -> concentration*
Changed.

- *- P3, L27 – I do not think PV is measured by the MLS.*
  You are right. We changed the text according to your suggestion.

- *- P3, L27 – proxy -> proxies, allow for transport -> allow transport*
  Changed.

- *- P3, L29-30 – Either add references here or explain how this can be done.*
  Beside the Asian monsoon e.g. typhoons can uplift air masses from the ground into the upper troposphere in the vicinity of the monsoon system. These air masses are dragged by the AMA and loop around the the outer part. A reference is added to the text.

- *- P3, L30 – regions are -> regions considered are*
  Not changed. Because this are the important source regions found in Vogel et al. 2016.

- *- P4, Table 1 – A map can be added here. Or this table can be replaced by a map.*
  We decided to skip the map, because it is already published in the Vogel et al. 2015 and Vogel et al. 2016 and would not provide more information than the table.

- *- P4, L3-4 – that these. . .stratosphere -> that these regions do not make a significant contribution to the lower stratosphere*
  Changed.

- *- P4, L7-8 – In contract to three -> In contrast to the three, temperature development -> changes in the temperature*
  Changed.

- *- P4, L8 – A reference is needed at the end of this sentence.*
  We mitigated the formulation of this sentence to: However, the position of an

air parcel and the changes in the temperature along the trajectory can be tracked with sufficient accuracy over several days or even weeks. In Addition, we included two studies of Vogel et al. 2014 and Rolf et al. 2015 which sucessfully used long trajectories.

- *- P4, L10 – remove 'of the ECMWF'*
  Changed.

- *- P4, L22 – 'the tropical Pacific' – I am not sure if this has been mentioned previously.*
  It is mentioned in the Section 2.3, where the contributing artifical tracer source regions are stated. Nothing changed here.

- *- P5, L1 – 'reinforce the hypotheses' – What does this mean?*
  We rephrased the text to make it more clear.

- *- P5, L6 – Figures 1a and b -> Figures 1a and 1b (also P9, L8)*
  Changed.

- *- P5, L7 – along equivalent latitude -> along the equivalent latitude. Also explain how 'equivalent latitude' is defined.*
  Changed and we added "PV based" as definition for equivalent latitude.

- *- P5, L8 – What does 'tropospheric influenced air' mean? Does this mean the air is mixed with tropospheric air?*
  Yes, exactly. In this case we mean stratospheric airmasses which are influenced by tropospheric airmasses by mixing. We rephrased the text to make it more clear.

- *- P5, L11 – have a similar. . .extent. . . -> have similar vertical and horizontal extents*
  Changed.

[Figure]

- *- P5, L16 – at a PV value -> at PV values, remove 'in the time'*
  Changed.

- *- P5, L19 – All air -> All the air*
  Changed.

- *- P5, L22 – The distribution -> The frequency distribution*
  Changed.

- *- P5, L25 – vapor concentration -> vapor concentration between the two phases*
  Changed.

- *- P5, L28 – It might be easier to refer a figure in Vogel at al., instead of a page number.*
  You are right, we changed the reference to the Figure instead of the pape.

- *- P5, L32 – Figures 2a, b -> Figures 2a and 2b*
  Changed.

- *- P5, L33 – remove 'relatively'*
  Changed.

- *- P6, Figure 1 – a)-d) can be written on the top left instead of bottom left (also in other figures).*
  Changed.

- *- P6, L3 – Figure 2d) -> Figure 2d*
  Changed.

- *- P6, L4 – from phase 2 to phase 1 -> from phase 1 to phase 2*
  Changed.

- - *P8, L2 – enhancements -> enhancement*
  Changed.

- - *P8, L5 – Only few -> Only a few*
  Changed.

- - *P8, L7 – Explain what 'core region' means.*
  The core region of the scatter plot is defined by the 75% percentile of the frequency distribution (black contour). It is now written in the text.

- - *P8, L10 – 'the slope of' – A linear regression line can be added in Fig. 3a.*
  We added a linear regression to the plot to better highlight the slope of the correlation.

- - *P8, L18 – What does 'the imprint on water vapor' mean here?*
  It is directly connected to the sentence before. "The amount of water vapor which is transported from the troposphere into the stratosphere is strongly coupled with the Lagrangian cold point (LCP), where typically the water vapor is dehydrated close to the saturation mixing ratio by ice crystal formation and subsequent sedimentation. Thus, the amount of water vapor in these air masses is nearly preserved after passing the LCP in the tropical, sub-tropical and mid-latitude stratosphere." The imprint means the minimum saturation mixing ratio along the temperature. We make it clear by changing the wording to "these imprint".

- - *P9, L1 – that Eddy shedding -> that the eddy shedding*
  Changed.

- - *P9, L5 – A reference can be added at the end of this sentence.*
  We added two references which address this point.

- - *P10, L2 – between both phases -> from phase 1 (Aug) to phase 2 (Sep)*
  Changed.

- - *P10, L7 – model study of -> modeling study of*
  Changed.

- - *P10, L20 – gives observational -> gives an observational*
  Changed.

[revised manuscript text omitted]

---

## Author Comment (AC2) · 9 Jan 2018

The authors thank the reviewer #2 for the many comments which were very helpful to improve the paper. All changes of the paper are highlighted in red color. Point by point answers to your comments are reported below.

**Major comments**

- *(1)My main concern is the novelty or substance of the paper. Moistening of the Ex-LS via the AMA has been reported before (Randel and Jensen, 2013), even*

[Figure]

*based on in-situ data from the same 2012 campaign and a similar methodology (Müller et al., 2016, Vogel et al., 2016). Vogel et al. (2016) also already quantified the moistening for the entire monsoon season of 2012. Although Rolf et al. provide additional and detailed analyses, new findings should be brought out better in the paper.*

We agree that the paper Vogel et al. (2016) / Müller et al. (2016) and our present study are strongly connected. However, the major contribution of the present study is the investigation of in-situ measurements of water vapor with a *high vertical resolution*. Clearly, vertical resolution is the key in UTLS processes and remote sensing instruments like MLS have clear deficits in this respect (Santee, JGR, 2017). Moreover, water vapor and changes in its distribution in the UTLS impacts the radiation budget and thus climate. Therefore, the knowledge about water vapor transport pathways, change rates, and sources is highly important. We added more information about this in the introduction.

Further, with this study we connect the in-situ measurement directly to the CLaMS model simulations and can partially confirm the moistening found by Vogel et al. 2016 with the model and with the MLS satellite data. Therefore, highly precise in-situ measurements are a justification and constitute a different method in estimating the moistening. The study by Müller et al., 2016 is focused in detailed on other trace gases like CO, O3, and N2O, which are important as well, but the study of Müller et al. 2016 is not complete in investigating transport pathways concerning water vapor.

**Methodology:**

- *(1) I don't really see the point in using tracers of airmass origin here. The tracer(s) described in the paper do not seem to be reliable indicators of AMA air, and do not provide additional relevant information as compared to back-trajectories.*
  It is important to note here that CLaMS transport is Lagrangian (i.e. by trajec-

tories) so that any CLaMS tracer (including air mass origin) will be transported mainly Lagrangian (i.e. by trajectories) with an *additional* impact of mixing. Thus, back-trajectories show "only" the history and the origin of an airmass with the underlying advection. The origin tracer are constantly emitted at the ground, are chemical inert, and are mixed in the atmosphere, which is an important difference to trajectories. Especially mixing processes like eddy shedding, which is one of the crucial transport pathways of monsoon air, can be better quantified. This is already shown by Vogel et al. 2015, 2016 where they showed that the origin tracers provide information of airmass composition of different source regions, which can be better associated with the true airmass composition compared to back-trajectories alone. In addition, with the combination of origin tracer we can better confirm the studies of Vogel et al. 2016 and Ploeger et al. 2013 and directly associate the hydration found in the measurements with the increase of airmass contribution from the Asia.

- *(2) The choice of the two time periods for detecting changes to H2O and CH4 seems to be rather arbitrary. The delta should be discussed in the context of the transport time scales involved, which might be available from back trajectories. Transport time scales also seem to be the key for relating the results of this paper to the numbers found by Vogel et al. (2016).*
  The periods and the two week break in between are the result of the campaign planning and are in this way arbitrary. But they fit perfectly to the period of the strongest increase in water vapor in the Ex-LS found by Vogel et al 2016 in CLaMS water vapor and in the MLS observation at 380 K and 400 K. Between the two phases, the strongest increase of Ex-LS water vapor is the result of a pronounced eddy-shedding event which occurred on 20 September 2012 (Vogel et al., 2014, 2016) and matches perfectly the separation of the two phases. We added more text in the discussion part (Sec. 3.1) and in the conclusion section of the paper, to relate the increase observed by Vogel et al. 2016 to the increase

found with the in-situ measurements.

- *(3) Convection is an important process when it comes to the AMA, but is notoriously hard to capture in large-scale models. Please discuss, how you consider convection.*

  Convection is the major processes to transport air masses vertically from the ground into the upper troposphere in the Asian monsoon region. In the model simulations shown here, vertical transport is taken into account as represented in ERA-Interim including latent heat related transport (for pressure less than $\approx 300$ hPa). Thus, the very small-scale rapid uplift in convective cores cannot included in CLaMS simulations. However, for the questions addressed here, which are concerned about large scale uplift in the monsoon, these small-scale features might be less relevant. In particular, previous studies using 3-D CLaMS simulations or trajectory calculations (e.g., Ploeger et al., 2010, 2015; Pommrich et al., 2014; Vogel et al., 2014, 2015; Müller et al., 2016; Ungermann et al., 2016; Konopka et al., 2016), in comparison with satellite or in situ measurements, show that ERA-Interim data are well suited for studying transport processes in the vicinity of the Asian monsoon anticyclone and in the tropical tropopause layer.

- *(4) The term "statistic" seems to have been added to the corresponding terms in response to the initial review, but I still do not see actual statistical calculations. Please provide details on the statistical methods used for those obviously non-Gaussian distributions, mark significant points/ranges where applicable, provide correlation coefficients, confidence intervals etc. This might not apply to each and every use of statistical terms (correlation, significance, . . .), but you should make clear where it is normal language rather than backed by maths.*

  You are right, this is an important information. We added correlation coefficients and p-values to show the significance at the important parts in the text. In the Section 3.1 we added: The difference in the mean water vapor mixing ratio between the two phases indicates a statistical significant (Mann-Whitney-U-Test

with p-value $< 10^{-100}$) moistening of the Ex-LS of about 0.5 ppmv within a time period of less than a month.

In Section 3.3 we provide correlation coefficients and further discuss them: The associated correlation coefficients for water vapor and methane are 0.79 and 0.67 for phase 1 and phase 2, respectively. The correlation coefficients for measured water vapor and modeled MON tracer are 0.27 in phase 1 and 0.40 in phase 2 and for methane and MON tracer 0.41 for phase 1 and 0.63 for phase 2. All correlation coefficient (calculated according to Pearson) have a very high significance with p-values of nearly zero ($< 10^{-18}$) rejecting the null hypothesis of an uncorrelated dataset. In fact, the correlation for water vapor and methane is obvious higher compared to the correlation with the MON tracer, which corroborates the consistency between both in-situ measurements. But also shows the limitations of CLaMS to reproduced every small scale feature in the measurements. However, the correlation for the model based tracer and the measurements is still quite satisfying and strongly supports the hypothesis that the increase in water vapor is correlated to airmass affected by the AMA.

- *(5) Flight paths determine the region of study and should be shown. Citing Müller et al. (2016) only helps partially, because they analyse slightly different periods.* We see the point and included a map with the flight paths and attribution to the respective phase into the manuscript.

- *(6) Please distinguish between "concentration" and "mixing ratio" throughout the paper. For instance, "ppbv" is a unit for mixing ratios (better: nmol/mol). The SI unit for molar concentration is mol/m3, for number concentration 1/m3.* Yes, that is totally right. We carefully scanned the manuscript and exchanged the wording according to the suggestion.

**Specific comments**
- *P1L8: Are those the exact numbers? Otherwise please use "about"*
  We added "about" in the text.

- *P1L13: exclusively -> mostly/predominantly*
  Changed to predominantly.

- *P2L9: AMA is leaky*
  We rephrased the text to make it more clear that the AMA constitute as more a weak transport barrier.

- *P2L29: ESMVal went around Africa*
  This is true. The campaign was conducted to get a full meridional cross section of the earth. So there were also measurements in the northern hemisphere. And for the for the total number of measurements the most part of both campaigns took place above Europe. Therefore we keep the current wording with "mainly over Europe and the Atlantic Ocean".

- *P2L30: Why those dates? The Monsoon starts in June/July, but your approach seems to be based on the assumption that less trajectories from the AMA have reached the flight paths at the beginning of Sep compared to the end of Sep. Please discuss the time scales for transport from the AMA to the measurements.*
  We choose this combination of dates because of the clear difference in their water vapor distribution in the Ex-LS. You are right, that the monsoon season starts earlier but concerning transport into the Ex-LS the later phase between August and September seems to be more important. This is also visible in Fig 15. of Vogel et al. 2016. The water vapor mixing ratio increases most strongly in the time between August and September. This study confirm the CLaMS modeling from Vogel et al. 2016 results quite nicely! We included this brief motivation for the selection if dates in the text.

- *P3L12: explain acronym*

Changed.

- *P3L28: Does that mean that this type of emission tracer preferentially ends up in the AMA? For using MON as a proxy for the AMA, it is not enough to show that MON is the main source of AMA air. Additionally, MON must not end up in significant amounts outside the AMA.*

  In this study, we focus on all airmasses affected by the AMA. This includes airmasses within the anticyclone as well as airmasses which are lifted to this altitudes and circulate aorund the AMA at the outer edge. Vogel et al. 2016 explains this in more detail: Maximum percentages for the emission tracer for south-east Asia/tropical Pacific Ocean are found at the edge of the Asian monsoon anticyclone, indicating the transport of these air masses around the outer edge of the Asian monsoon anticyclone outside of a PV-based transport barrier (Ploeger et al., 2015) as discussed by Vogel et al. (2015). Thus, air masses from southeast Asia/tropical Pacific Ocean are found within a widespread area around the anticyclone caused by the large-scale anticyclonic flow in this region acting as a large-scale stirrer. Moreover, large contributions of these emission tracers are found in the tropics associated with deep uplift outside of the monsoon region.

  Also note that the composition of the AMA is not static, rather the composition is influenced by a steady upward flux of boundary layer air, which leads to a different composition of the AMA air over the monsoon period (see Fig. 8 in Vogel et al. 2015 and the related discussion).

- *P4L17: Does possible supersaturation play a role for these analyses?*

  Supersaturation is needed to initiate ice formation, but is reduced to the saturation mixing ratio within short time. This occurs typically at the coldest point along the trajectory. With additional ice crystal sedimentation the water vapor mixing ratio ends up at the saturation mixing ratio without ice in a first order approximation. Especially, with the long transport times of several days in the lower stratosphere

supersaturation and ice formation don't play a role. Thus, for this type of analysis we don't see a need to account for supersaturation.

- *P4L20: Consider rewording to make clear that both conditions must be met.*
  It is now rephrased.

- *P5L9: The motivation for choosing potential temperature difference to the local ther- mal tropopause is not clear to me. Water vapour depends on temperature, but the thermal tropopause is based on a lapse rate. Also, potential temperature is good for characterizing isentropic transport, but not necessarily temperature history.*
  The thermal tropospause in the sub-tropics and mid-latitude is found to be a good indicator to mark the critical level of temperature gradient and sharp relative humidity change as shown by e.g. Pan and Munchak 2011. Thus separates the moist tropospheric regime from the dry stratospheric regime. Therefore the thermal tropospause seems to be a good estimate concerning the water vapor gradient. It is true, that isentrophes do not show the temperature history, but we decided to use this coordinate mostly associate the water vapor distribution to the underlying isentropic transport. We added this information to the text to make it clearer.

- *P5L10: Please briefly motivate using equivalent latitude*
  We included a short motivation in the text.

- *P5L11: consider choosing the contours in Fig. 1 to display the threshold used in the text*
  We tried to put contour lines into these figures, but they are getting confusing and messy because the water vapor distribution is not that smooth. Therefore we decided to skip additional contour lines in these plots. It is better visible with just the colors from the colorbar.

- *P5L15: Fig. 1 only shows equivalent latitude as vertical coordinate. Were similar longitudinal regions sampled during both phases?*
  We show equivalent latitude as horizontal coordinate. The most data were collected in an longitude range between 15W and 20E during both phases (phase1 86% and phase2 75% of the measurements). In phase 1 also measurements more in westerly direction up to 24W and in phase 2 more in easterly direction up to 60E where conducted. So we think that both phases are comparable. We also think that the exact longitudinal coverage is of minor importance because of the variability in the Ex-LS at least above 8 PVU is low and the increase of water vapor in the Ex-LS is a general feature in the entire northern Ex-LS in this time period.

- *P5L16: This sentence is not clear to me. Do you mean that below 8PVU some "pixels" in your eqLat-DTTP-space show larger differences than others of H2O between the two phases? How does the thermal tropopause affect local variability? Fig. 1c shows local differences in water vapour. It sounds a bit odd to attribute local differences of water vapour to local variability of water vapour. Consider revising.*
  The thermal tropopause used here is derived from ECMWF data and does not always match perfectly the true tropopause height. In this case you would attribute moist tropospheric air masses to the stratospheric regime and vice versa. So in the end, it is a mismatch of the applied criterion directly around the tropospause. You are right, that it is a odd argument to attribute local differences of water vapor to local variability of water vapor. We revised the text concerning this point.

- *P5L26: You point out that the distribution is compact for period 1. This gives the impression that most of the moistening happens between the two periods. However, period2 is already at the end of the monsoon season. Why is 0.5ppmv consistent with 1-1.5ppmv then? What are the transport time scales involved?*
  The moistening begins already in the beginning of July and continue until beginning of October. The gradient of increasing water vapor mixing ratio is strongest between both phase. This is already shown by Vogel et al. 2016. and clearly visible in Figure 15, which shows the increase of water vapor from Clams as well as MLS satellite measurements. In fact, our high precise in-situ measurements are a strong confirmation of the MLS and CLaMS results. We included one additional sentence to make the connection between both papers more clear.

- *P5L28: How did you test significance for those non-Gaussian distributions?*
  We tested the difference between bot distributions with the Mann-Whitney-U-Test. We found very small p-values of 1e-120 which indicates a significant difference between both distribution. This we did also for the monsoon tracer and for the methane distribution and found similar results.

- *P5L28: Are your measurements representative of the entire Ex-LS?*
  The measurements during TACTS and ESMVAL were focused both to get a meridional cross-section at the subtropical jet and from equator to the pole. The flight paths were not planed to catch explicitly air from the Asian monsoon, but rather to get a climatological representation of the Ex-LS. Also the agreement of between CLams and MLS, mentioned in the previous comment and the correlation of Asian monsoon surface tracer indicate the representativeness of the water vapor measurements presented in this study.

- *P5L33: Please show CH4. It might support your claim that the signature originates in the AMA.*
  We included the PDF for both phases of methane for PV larger than 8 in the manuscript. In addition, we added also the correlation coefficient for water vapor and methane in the text to corroborate the increase of water vapor to the AMA.

- *P6L4: 100*
  We are not sure what is meant here.

- *P6L5: Do you just want to say increasing H2O corresponds to increasing MON? How- ever, providing some correlation coefficients for the region above 8PVU would be good. To me the pattern above 8PVU looks different between Figs. 1c and 2c.*

  The pattern of Fig. 1c and 2c does not match perfectly, this is correct. We have to keep in mind however that we compare here a model with measurements, which includes always some deviations. The correlation coefficients for water vapor and MON tracer are 0.27 in phase1 and 0.40 in phase2. The correlation coefficient for methane and MON tracer are 0.41 for phase1 and 0.63 for phase2. All correlation coefficient have a very high significance with p-values of nearly zero ($< 1e$-20). In fact, methane shows a better correlation. This is because methane has a long life time of around 8.9 years in the troposphere and even longer in the lower stratosphere (Wuebbles et al. 2002), whereas water vapor has a very short life time of about 9 days (Hartmann 2015) due to its ability to condense and change phase in response to temperature. Therefore we can expect a more reduced variability of water vapor in comparison to methane, due to the damping effect of the Lagrangian cold point (LCP). This reduces the correlation in the case of water vapor. This is also the reason, why we performed the trajectory analysis concerning the LCP to show the origin of the significant increase in the water vapor concentration. We included the numbers and part of this explanation in Sec. 3.3, were the correlations of water vapor, methane and MON tracer are discussed.

- *P7L8: "Thus" and "because" in one sentence is confusing. Please clearly ex- press why you consider CH4 to be a monsoon tracer. Please provide references for each argument.*

  The text is now rephrased.

- *P8L4: What is the statistically significant region in this plot? Please provide numbers for the background mixing ratios.*

The statistical significant region are represented by the water vapor mixing ratio below the background and show that the influence of air masses from the AMA is of minor importance for the first phase. We included also the background mixing ratio of 5 ppmv in text, which fits to the limits of the core region.

- *P8L5: The grammar of this sentence is odd. However, no matter how I interpret it, the definition of the core region remains unclear. If the core region is defined by the 75*
  We rephrased the text to make it clearer.

- *P8L11: sounds odd, consider rewording*
  Rephrased.

- *P8L12: This sentence is not clear to me. Did you test statistical significance? If so: How exactly did you do that? The scatter plots show H2O mixing ratio versus CH4 mixing ratio. If you want to discuss the statistics of deltas ("increase"), please show deltas.*
  You are right. We tested the methane and water vapor concentration to the MON tracer and not directly the increase. We changed the text and show some correlation coefficient and an explanation.

- *P8L12: slope CH4/H2O -> tilted towards a higher ratio or higher CH4, but not towards higher H2O. Please revise. Also, you discussed that contributions from the ASM in- crease H2O and CH4. Fig. 3 looks like the ratio CH4/H2O changes. Is that also a tell-tale sign of ASM origin?*
  We revised the text according to your suggestion. We also recognize the change in the ratio and found with the help of the emission tracer the slope is dependent on the source region. Low methane high water vapor seems to come from the tropical Pacific, whereas the high methane in phase 2 is more from the South east Asia. But we found this result to be to speculative, therefore we excluded it from the text.

- *P8L17: 1. "Water vapor is dehydrated" sounds odd; 2. Do you mean that dehydration typically happens close to the saturation mixing ratio, or that air masses typically get close to saturation at the LCP? (Only wet air might produce ice crystals). Please revise.*
  That's right. We revised the text. Ice crystals can only form in supersaturation therefore it is unlikely that air masses can be moister than the LCP. It can happen, if not all ice crystals sediment out of the air mass. The LCP is also the coldest point along the airmass history and therefore it is unlikely that you find dryer tropospheric air masses than the saturation mixing ratio at the LCP.

- *P9L2: Are you considering only trajectories that originate in the troposphere here? The LCP that determined water vapor might lie further back than the reach of the backward trajectory.*
  Yes, we took only trajectories, which were in the troposphere during the last 50 days. It might be that the LCP for some of the stratospheric trajectories lie further back in time, but these trajectories cannot be related to the AMA. This is just because the it would be before the onset of the Asian monsoon. Other mixing processes seems to be week and unimportant for water vapor, because in the first phase we observe most mixing ratios up to the stratospheric background.

- *P9L10: remove "notably" or "interestingly"*
  Changed.

- *P9L20: ... and region of the flights?*
  We clam, that the whole northern Ex-LS is affected by this water vapor enhancement and this includes of course also the region of the flights.

- *P9L23: Fig. 5 can be removed. The next sentence contains all the information.*
  You are right, that all information can also be found in the text. However, we decided to keep the Figure because it supports the reader to better recognize the

proportions of tropospheric and stratospheric air masses within the two phases in comparison to numbers in the text only.

- *P9L25: amount -> fraction or percentage*
  Changed to fraction.

- *P9L25: in phase -> contributing to phase (Please have in mind that transport from the AMA needs some time.)*
  This is true. We changed the text.

- *P10L5: How did you calculate the correlation? Please provide numbers.*
  This is true, we did not calculated the statistical significance of the increase itself. We calculated the correlation between the different tracers. We revised the text to clarify this point.

- *P10L6: distinguish between AMA and the respective surface tracers*
  We changed the text to "region of the Asian monsoon". This is more specific in terms of the origin tracers.

- *P10L11: I am not convinced. Please elaborate on how you relate your results to Vogel et al. (2016). Is water vapour transport from the AMA to the ex-LS constant throughout the monsoon season? How do you relate your phase1/gap/phase2 periods to the monsoon period? Are the diagnostics comparable?*
  See answer to Methodology part 2. We included also more explanation in the conclusion section.

- *P10L19: crossing is displaced OR crossings are displaced; What is combined with quasi-isentropic transport? Please revise and consider splitting it into two sentences.*
  We revised the text according to your suggestions.

- *P10L21: "picture of influencing" sounds odd*
  Changed.

- *P10L25: This should be shown, at least via a rough estimate.*
  This is a very good suggestion, but beyond the scope of this paper. There are several studies showing the climatic impact of water vapor changes in the UT/LS. For example Riese et al. 2012 showed that a 5-10% increase of water vapor in the Ex-LS produced a top of the atmosphere radiative forcing of 100-500 mWm-2. We included this information also in the conclusion.

**References:**

- Hartmann, Dennis: Global Physical Climatology (Second Edition), Elsevir, ISBN: 978-0-12-328531-7,p. 399, 2015.

- Konopka, P., F. Ploeger, M. Tao, and M. Riese (2016), Zonally resolved impact of ENSO on the stratospheric circulation and water vapor entry values, J. Geophys. Res. Atmos., 121, 11,486–11,501, doi:10.1002/2015JD024698.

- Müller, S., Hoor, P., Bozem, H., Gute, E., Vogel, B., Zahn, A., Bönisch, H., Keber, T., Krämer, M., Rolf, C., Riese, M., Schlager, H., and Engel, A.: Impact of the Asian monsoon on the extratropical lower stratosphere: trace gas observations during TACTS over Europe 2012, Atmospheric Chemistry and Physics, 16, 10 573–10 589, https://doi.org/10.5194/acp-16-10573-2016, 2016.

- Pan, L. L., and L. A. Munchak (2011), Relationship of cloud top to the tropopause and jet structure from CALIPSO data, J. Geophys. Res., 116, D12201, doi:10.1029/2010JD015462.

- Ploeger, F., Konopka, P., Günther, G., Grooß, J.-U. and Müller, R.: Impact of the vertical velocity scheme on modeling transport in the tropical tropopause layer.

Journal of Geophysical Research 115: doi: 10.1029/2009JD012023. issn: 0148-0227, 2010.

- Ploeger, F., Günther, G., Konopka, P., Fueglistaler, S., Müller, R., Hoppe, C., Kunz, A., Spang, R., Grooß, J. U., and Riese, M.: Horizontal water vapor transport in the lower stratosphere from subtropics to high latitudes during boreal summer, Journal of Geophysical Research- atmospheres, 118, 8111–8127, https://doi.org/10.1002/jgrd.50636, 2013.

- Ploeger, F., Gottschling, C., Griessbach, S., Grooß, J. U., Günther, G., Konopka, P., Müller, R., Riese, M., Stroh, F., Tao, M., Ungermann, J., Vogel, B., and von Hobe, M.: A potential vorticity-based determination of the transport barrier in the Asian summer monsoon anticyclone, Atmospheric Chemistry and Physics, 15, 13 145–13 159, https://doi.org/10.5194/acp-15-13145-2015, 2015.

- Pommrich, R., Müller, R., Grooß, J. U., Konopka, P., Ploeger, F., Vogel, B., Tao, M., Hoppe, C. M., Günther, G., Spelten, N., Hoffmann, L., Pumphrey, H. C., Viciani, S., D'Amato, F., Volk, C. M., Hoor, P., Schlager, H., and Riese, M.: Tropical troposphere to stratosphere transport of carbon monoxide and long-lived trace species in the Chemical Lagrangian Model of the Stratosphere (CLaMS), Geoscientific Model Development, 7, 2895–2916, https://doi.org/10.5194/gmd-7-2895-2014, 2014.

- Randel, W. J. and Jensen, E. J.: Physical processes in the tropical tropopause layer and their roles in a changing climate, Nature Geoscience, 6, 169–176, https://doi.org/10.1038/ngeo1733, 2013.

- Santee, M. L., Manney, G. L., Livesey, N. J., Schwartz, M. J., Neu, J. L., and Read, W. G.: A comprehensive overview of the climatological composition of the Asian summer monsoon anticyclone based on 10 years of

[Figure]

Aura Microwave Limb Sounder measurements, Journal of Geophysical Research: Atmospheres, 122, 5491–5514, https://doi.org/10.1002/2016JD026408, http://dx.doi.org/10.1002/2016JD026408, 2016JD026408, 2017.

- Vogel, B., Günther, G., Müüller, R., Grooß, J. U., Afchine, A., Bozem, H., Hoor, P., Krämer, M., Müller, S., Riese, M., Rolf, C., Spel- ten, N., Stiller, G. P., Ungermann, J., and Zahn, A.: Long-range transport pathways of tropospheric source gases originating in Asia into the northern lower stratosphere during the Asian monsoon season 2012, Atmospheric Chemistry and Physics, 16, 15 301–15 325, https://doi.org/10.5194/acp-16-15301-2016, 2016.

- Vogel, B., Günther, G., Müller, R., Grooß, J. U., and Riese, M.: Impact of different Asian source regions on the composition of the Asian monsoon anticyclone and of the extratropical lowermost stratosphere, Atmospheric Chemistry and Physics, 15, 13 699–13 716, https://doi.org/10.5194/acp-15-13699-2015, 2015.

- Vogel, B., Günther, G., Müller, R., Grooß, J. U., Hoor, P., Krämer, M., Müller, S., Zahn, A., and Riese, M.: Fast transport from Southeast Asia boundary layer sources to northern Europe: rapid uplift in typhoons and eastward eddy shedding of the Asian monsoon anticyclone, Atmospheric Chemistry and Physics, 14, 12 745–12 762, https://doi.org/10.5194/acp-14-12745-2014, 2014.

- Ungermann, J., Ern, M., Kaufmann, M., Muller, R., Spang, R., Ploeger, F., Vogel, B., and Riese, M.: Observations of PAN and its con- finement in the Asian summer monsoon anticyclone in high spatial resolution, Atmospheric Chemistry and Physics, 16, 8389–8403, https://doi.org/10.5194/acp-16-8389-2016, 2016.

- Wuebbles, D. J. and Hayhoe, K.: Atmospheric methane and global change, Earth-science Reviews, 57, PII S0012–8252(01)00 062–9, https://doi.org/10.1016/S0012-8252(01)00062-9, 2002.